# Decoding the Human Epidermal Complexity at Single-Cell Resolution

**DOI:** 10.3390/ijms24108544

**Published:** 2023-05-10

**Authors:** Maria Pia Polito, Grazia Marini, Michele Palamenghi, Elena Enzo

**Affiliations:** Centre for Regenerative Medicine ‘‘Stefano Ferrari’’, University of Modena and Reggio Emilia, 41125 Modena, Italy

**Keywords:** human keratinocyte, stem cell, single-cell analysis, skin, inflammatory disease, wound healing

## Abstract

The epidermis is one of the largest tissues in the human body, serving as a protective barrier. The basal layer of the epidermis, which consists of epithelial stem cells and transient amplifying progenitors, represents its proliferative compartment. As keratinocytes migrate from the basal layer to the skin surface, they exit the cell cycle and initiate terminal differentiation, ultimately generating the suprabasal epidermal layers. A deeper understanding of the molecular mechanisms and pathways driving keratinocytes’ organization and regeneration is essential for successful therapeutic approaches. Single-cell techniques are valuable tools for studying molecular heterogeneity. The high-resolution characterization obtained with these technologies has identified disease-specific drivers and new therapeutic targets, further promoting the advancement of personalized therapies. This review summarizes the latest findings on the transcriptomic and epigenetic profiling of human epidermal cells, analyzed from human biopsy or after in vitro cultivation, focusing on physiological, wound healing, and inflammatory skin conditions.

## 1. Introduction

The human epidermis, the outermost layer of the skin, is composed of stratified squamous epithelium that acts as a protective barrier against external insults; hence, any alteration can drastically impair patients’ life quality [1,2,3].

The basal layer represents the proliferative compartment of the epidermis, in which both epithelial stem cells (SCs) and transient amplifying progenitors (TACs) reside. As keratinocytes leave the basal layer and move towards the skin surface, they exit the cell cycle and undergo a terminal differentiation program, generating the suprabasal epidermal layers (spinosum, granulosum, lucidum, and corneum) [4,5]. This process, known as cornification or keratinization, consists of morphologic and metabolic changes whose endpoint is to form the cornified layer or the stratum corneum, the outermost layer of the epidermis [6].

In the last four decades, the ex vivo restoration of functional epithelia allowed the treatment of different medical needs [7,8,9,10]. An astonishing example has been the life-saving treatment of third-degree burn and junctional epidermolysis bullosa (JEB)-affected patients [10] thanks to the knowledge acquired during the studies conducted by H. Green in the 1980s [7]. The successful outcome of therapeutic approaches relies on a fine understanding of the molecular mechanisms and pathways driving human keratinocytes’ organization and regeneration. In this context, single-cell technologies have already and will further increase our knowledge on in vivo and in vitro cellular complexity.

Single-cell RNA sequencing (scRNA-seq), unlike bulk analysis, allows researchers to decipher the complexity of biological systems at the single-cell level. In 2006, a pioneering study involving the transcriptomic profiling of 314 single manually picked keratinocytes enabled the identification of 6 SCs and 6 TACs. Despite the low sensitivity, this work paved the way for an in-depth analysis of keratinocyte molecular heterogeneity [11].

The introduction of microfluidic systems for cell encapsulation drastically reduced time and effort, thus increasing reproducibility to study the transcriptome of thousands of cells [12,13]. Two main library construction methods have been developed: full-length methods (e.g., SMART-seq2, Fluidigm C1) cover the entire transcriptome, while molecular tag-based methods (e.g., 10× Genomics Chromium, MARS-seq, InDrop, Drop-seq) analyze the mRNA 5′ or 3′ ends. Molecular tag-based methods allow for a large cell throughput and sample multiplexing to improve gene expression quantification, whereas full-length methods display higher sensitivity [14,15]. An in-depth review of single-cell approaches can be found elsewhere [16,17,18].

The high-resolution map obtained through these technologies opened new scenarios, allowing the identification of new cell types, disease-specific drivers, and new therapeutic targets, further promoting the advancement of personalized therapies. To allow a broader diffusion of these data in the scientific community, atlases of single-cell-derived data are now publicly available, such as the Skin Community of Human Cell Atlas (HCA, https://skincommunity.org/hca, accessed on 6 May 2023 [19]) or Adult Human Cell Atlas (AHCA, http://research.gzsums.net:8888/, accessed on 6 May 2023 [20]).

In this review, we summarize the latest characterization of human epidermal cells using single-cell technologies, which could help to develop and improve innovative therapeutic approaches, focusing on skin regeneration therapies, wound care, and inflammatory diseases.

## 2. Skin Biopsy Processing

Given the considerable skin-extension and site-specific functions, experimental settings (in vivo or in vitro) (Table 1), biopsy withdrawal and processing methods (Table 1), the anatomical areas of collection (Figure 1), and the age and pathology of the donors (Figure 2) must be considered to properly analyze single cells’ published data.

The less invasive and more common biopsy techniques for research purposes are shaved and punch biopsies, which enable the collection of samples from both the epidermis and the underlying dermis. Depending on the cell populations of interest, a skin biopsy can be processed in different ways: heat, chemical reagents, enzymes, or mechanical digestion. However, heat treatment may cause thermal damage, and chemical reagents may alter cellular electrolyte equilibrium. Hence, the most used methods rely on enzymatic activity or mechanical separation to divide the epidermis from the dermis [21]. Several proteases (dispases, trypsins, pancreatin, pronase, and thermolysin) have been tested to separate different layers of the dermal–epidermal junction [21,22].

A clinically validated method specifically used for the in vitro culture of human primary keratinocytes consists of a biopsy cleaning step to remove the adipose tissue and partially the dermis. Then, the tissue is minced and incubated in a trypsinizing flask with a trypsin/EDTA solution (mixture of trypsin, chymotrypsin, and elastase) for 30 min at 37 °C. Cells are recovered after each round of trypsinization and plated onto a lethally irradiated feeder layer [23]. This procedure allows the collection of both interfollicular and follicular keratinocytes, including SCs used for cell and gene therapy applications (unpublished data and [10,24,25]).

Another popular method takes advantage of dispase I, one of the most used enzymes to gently separate the dermis from the epidermal layer. It acts on extracellular matrix (ECM) proteins, including fibronectin, collagen IV, and to a lesser extent collagen I, allowing the collection of the entire epidermis [22,26]. Then, trypsin/EDTA must be used to gather single-fibroblast or single-keratinocyte suspensions [27]. Notably, most cell types display a distinctive stress signature related to the dissociation step, which must therefore be taken into account for further downstream analysis [28].

Suction blistering is another approach used to collect skin cells while avoiding extensive enzymatic digestion. High-vacuum mechanical forces are applied to the patient skin in this in vivo procedure, allowing the collection of epidermal and upper dermal cells [29,30]. Recently, a scRNA-seq comparison between suction blistering and normal punch biopsies has been reported. Some cell types were under-represented in suction blistering biopsies; however, both sampling techniques shared most of the identified pathways [30]. In addition, the authors claimed that suction blistering led to a better transcriptomic resolution of skin cells, also presenting the possibility to combine interstitial fluid analysis at the protein level [29,30].

**Figure 1 ijms-24-08544-f001:**
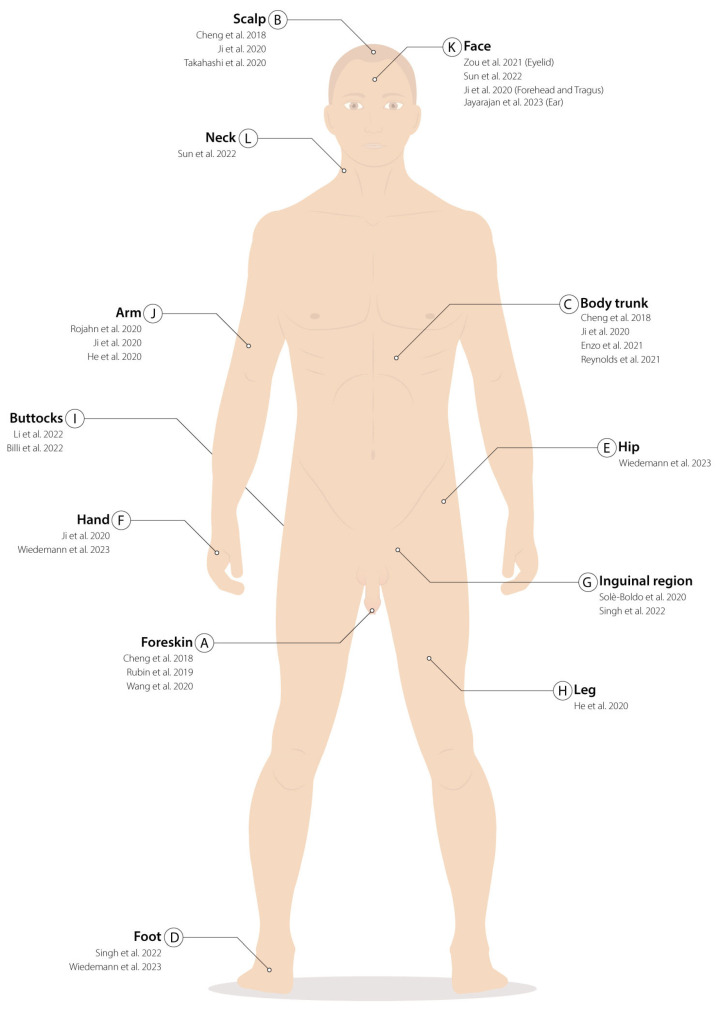
Biopsy withdrawal localizations: (**A**–**L**) Areas where skin cells have been harvested to be analyzed with single-cell techniques. For each location, references to the papers that performed the biopsies are listed in chronological order. (**A**) Foreskin [31,32,33]; (**B**) Scalp [31,34,35]; (**C**) Body trunk [25,31,34,36]; (**D**) foot [37,38]; (**E**) hip [38]; (**F**) hand [34,38]; (**G**) inguinal region [37,39]; (**H**) leg [40]; (**I**) buttocks [41,42]; (**J**) arm [30,34,40]; (**K**) face (forehead, eyelid, and tragus) [34,43,44,45]; (**L**) neck [43].

**Table 1 ijms-24-08544-t001:** Schematic description of single-cell techniques adopted, area and mode of biopsy processing, type of tissue analyzed, and data availability for each article cited in this review. SCT: single-cell techniques; scRNA-seq: single-cell RNA sequencing; scATAC-seq: single-cell Assay for transposase-accessible chromatin sequencing; ST: spatial transcriptomics; AW: acute wound; FACS: fluorescence-activated cell sorting; n/a: not available.

Article	SCT	Single-Cell Technology	Source	Conditions	In Vivo/In Vitro	Biopsy Processing Protocol	Data Availability
Reynolds et al. Science. 2021 [25]	scRNA-seq	10× Genomics Chromium + SMART-seq2	Trunk Lower back Embryonic/fetal skin from Popescu DM et al., 2019	Healthy Inflammatory	in vivo	Sample mincing 2U/mL dispase II: 1–3 h, 37 °C 1.6 mg/mL type IV collagenase: 12 h, 37 °C (both dermis and epidermis separately)	E-MTAB-8142
Gellatly et al. Sci Transl Med. 2021 [29]	scRNA-seq	In-Drop	Skin (not specified)	Healthy Inflammatory	in vivo	Suction blistering Negative pressure (10–15 mm Hg): 30–60 min, 40 °C Collection via an insulin syringe	phs002455.v1.p1
Rojahn et al. J Allergy Clin Immunol. 2020 [30]	scRNA-seq	10× Genomics Chromium	Antecubital fossa skin (not specified)	Healthy Inflammatory	in vivo	Suction blistering: Negative pressure (150–200 mm Hg), 1 to 2 h 0.25% trypsin/EDTA: 10 min, 37 °C (blister roof) FACS sorting Biopsies: Sample mincing 0.3% *w*/*v* collagenase IV, 40 min, 37 °C 40 µm cell-strainer 0.25% trypsin/EDTA, 10 min, 37 °C FACS sorting	GSE153760
Cheng et al. Cell Rep. 2018 [31]	scRNA-seq	10× Genomics Chromium	Foreskin Trunk Scalp	Healthy Inflammatory	in vivo	Dispase: 2 h, 37 °C Trypsin: 15 min, 37 °C (only epidermis) 40-μM cell strainer	EGAS00001002927
Rubin et al. Cell. 2019 [32]	scRNA-seq	Fluidigm C1	Foreskin	Healthy	in vitro	n/a	GSE116297
Wang et al. Nat Commun. 2020 [33]	scRNA-seq ST	10× Genomics Chromium	Foreskin	Healthy	in vivo	Dispase: 2 h, 37 °C Sample mincing 0.25% Trypsin-EDTA: 15 min, 37 °C (only epidermis) 40 µm cell-strainer	GSE147482
Ji et al. Cell. 2020 [34]	scRNA-seq	10× Genomics Chromium + 10× Genomics Visium	Dorsal hand Arm Scalp Trunk Forehead Tragus	Healthy Cancer	in vivo	Sample mincing 0.25% trypsin-EDTA: 30 min, 37 °C	GSE144240
Takahashi et al. J Invest Dermatol. 2020 [35]	scRNA-seq	DropSeq + 10× Genomics Chromium	Scalp	Healthy	in vivo	Dispase: overnight, 4 °C + 30 min, 37 °C P1000 pipette gentle dissociation (×2) 0.05% trypsin: 10 min, 37 °C (×2) 40 μM cell strainer FACS sorting	GSE129611
Enzo et al. Nat Commun. 2021 [36]	scRNA-seq	10× Genomics Chromium	Trunk	Healthy	in vitro	0.05% trypsin/0.01%EDTA: 4 h, 37 °C Sample collection every 30 min	GSE155817
Singh et al. J Clin Invest. 2022 [37]	scRNA-seq ST	10× Genomics Chromium + 10× Genomics Visium	Inguinal region Heel Skin (not specified) AW and intact skin from Li et al., 2022	Healthy	in vivo	Sample mincing MACS whole-skin dissociation kit: 3 h, 37 °C GentleMACS™ Octo dissociator: Skin dissociation program 70 μM cell strainer	GSE176417
Wiedemann et al. Cell Rep. 2023 [38]	scRNA-seq	10× Genomics Chromium	Sole Palm Hip	Healthy	in vivo	0.4% dispase: overnight, 4 °C 0.25% Trypsin-EDTA (+10 U/mL DNase I): 1 h, 37 °C (only epidermis) 70 μM cell strainer	GSE202352
Solè-Boldo et al. Commun Biol. 2020 [39]	scRNA-seq	10× Genomics Chromium	Inguinoiliac region	Healthy Aging	in vivo	Whole-skin dissociation kit for human material and Gentle MACS dissociator from Miltenyi Biotec	GSE130973
He et al. J Allergy Clin Immunol. 2020 [40]	scRNA-seq	10× Genomics Chromium	Body extremities	Healthy Inflammatory	in vivo	Biopsy specimens were cryopreserved, dissociated, and processed by 10× Genomics	GSE147424
Li et al. J Invest Dermatol. 2022 [41]	scRNA-seq	SMART-seq2	Chronic-wounded Skin Buttock	Healthy	in vivo	5 U/mL Dispase II: overnight, 4 °C 0.025% trypsin/EDTA: 10 min, 37 °C	GSE137897
Billi et al. Sci Transl Med.2022 [42]	scRNA-seq ST	10× Genomics Chromium + 10× Genomics Visium	Skin (not specified) Buttock	Healthy Inflammatory	in vivo	0.4% dispase: overnight, 4 °C. Epidermis and dermis were separated 0.25% trypsin-EDTA (+ deoxyribonuclease I): 1 h, 37 °C (only epidermis) Sample mincing (only dermis) 0.2% collagenase II-0.2% collagenase V: 1.5 h, 37 °C (only dermis) 70 μM cell strainer, 1:1 ratio	GSE186476
Sun et al. Front Cell Dev Biol. 2022 [43]	scRNA-seq	10× Genomics Chromium	Face Neck	Healthy	in vivo	2 mg/mL dispase II: overnight, 37 °C trypsin-versene: 10 min, 37 °C (115 rpm)	PRJNA797897
Zou et al. Dev Cell. 2021 [44]	scRNA-seq	10× Genomics Chromium	Eyelid	Healthy Aging	in vivo	Sample mincing 2 mg/mL collagenase I, 2 mg/mL collagenase IV, 2 mg/mL dispase, and 0.125% trypsin-EDTA: 1 h, 37 °C, 40 μM cell strainer	HRA000395
Jayarajan et al. Cells. 2023 [45]	scRNA-seq	10× Genomics Chromium	Plastic surgery for ear reconstruction	Healthy	in vitro	Sample mincing 0.02 U/mL neutral protease: 3 h, 37 °C 0.25% trypsin\0.01% EDTA: 5 min, 37 °C (only epidermis)	GSE207130
Schäbitz et al. Nat Commun. 2022 [46]	scRNA-seq ST	10× Genomics Chromium + 10× Genomics Visium	Skin (not specified)	Healthy Inflammatory	in vivo	MACS whole skin dissociation kit: 3 h, 37 °C	GSE206391
Harirchian et al. J Invest Dermatol. 2019 [47]	scRNA-seq	10× Genomics Chromium	Skin (not specified)	Healthy Inflammatory	in vivo	25 U/mL dispase: overnight, 4 °C 0.03% Trypsin: 15 min at 37 °C (only epidermis) 100-µm cell strainer	EGAS00001002981
Der et al. JCI Insight. 2017 [48]	scRNA-seq	Fluidigm C1	Skin (not specified) Kidney	Inflammatory	in vivo	0.25 mg/mL Liberase: 15 min, 37 °C 0.03% Trypsin: 10 min, 37 °C 70-μm cell strainer	PRJNA379992
Guerrero-Juarez et al. Sci Adv. 2022 [49]	scRNA-seq	10× Genomics Chromium	Skin (not specified)	Healthy Cancer	in vivo	dispase II-collagenase IV: overnight, 4 °C 0.25% trypsin-EDTA: 15 min, 37 °C for 15 min at 37 °C 40 μM cell strainer FACS sorting	GSE141526

**Figure 2 ijms-24-08544-f002:**
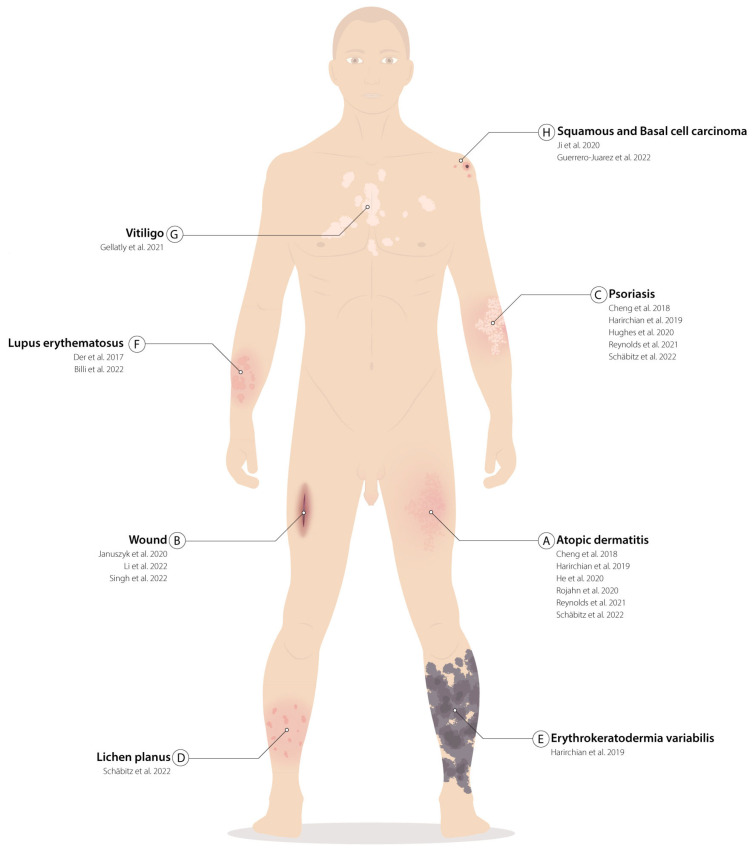
Skin pathological tissues analyzed in single cells: (**A**–**H**) List of skin diseases that have been characterized with single-cell techniques. For each disease, references to the papers that have characterized the specimens are listed in chronological order. Localization of the illustrated pathologies does not reflect the actual position from which the biopsies were withdrawn but serves as an example. (**A**) Atopic dermatitis [20,25,30,31,46,47]; (**B**) Wound [37,41,50]; (**C**) Psoriasis [25,31,46,47,51]; (**D**) Lichen planus [46]; (**E**) Erythrokeratodermis variabilis [47]; (**F**) Lupus erythematosus [42,48]; (**G**) Vitiligo [29]; (**H**) Squamous and basal cell carcinoma [34,49].

## 3. Single-Cell Molecular Profiling of In Vivo Epidermis

Skin diseases can be ascribed to alterations in both basal or suprabasal keratinocytes, to other epidermal cells, or to cell–cell communication networks established in vivo between the different skin layers. In light of this, the single-cell analysis of the entire skin biopsy could be relevant to better understand both healthy and pathological conditions.

### 3.1. Deciphering Complexity of Healthy Human Epidermis Using Single-Cell Approaches

Foreskin is one of the best-characterized epithelia, having an high proliferative potential (Figure 1A) [31,32,33]. However, data of skin collected from several anatomical areas, such as the scalp (Figure 1B) [31,34,35]; the truncal skin (Figure 1C) [25,31,34,36]; the foot (Figure 1D) [37,38]; the hip (Figure 1E) [38]; the hand (Figure 1F) [34,38]; the inguinal region (Figure 1G) [37,39]; the leg (Figure 1H) [40]; the buttock (Figure 1I) [41,42]; the arm (Figure 1J) [30,34,40]; the face (forehead, eyelid, and tragus) (Figure 1K) [34,43,44,45]; and the neck (Figure 1L) [43] are present in literature. Most of the studies on the human epidermis involve the interfollicular epidermis. Hair follicles have only been studied marginally in scalp biopsies [31], or from discarded human scalp micrografts collected for transplantation [35] (Figure 1D).

The first scRNA-seq report of human keratinocyte heterogeneity was published by Cheng et al., who studied epithelial complexity in different anatomical sites by collecting nine biopsies among foreskin, scalp, and truncal skin (Table 1, Figure 1A–C) [31]. They found 11 main clusters, 8 of which were identified as keratinocytes, 2 as melanocytes, and 1 composed of immune cells. Among the eight keratinocyte clusters, they distinguished two basal clusters expressing *KRT14* and *KRT5*, one formed by spinous keratinocytes expressing *KRT1* and *KRT10* and one granular cluster marked by *FLG* and *LOR*. In addition, they identified four other clusters defined as WNT1 (*SFRP1*^+^ and *FRZB*^+^); follicular (*MGST1*^+^ and *S100A2*^+^); mitotic (*CCND1*^+^ and *PCNA*^+^); and channel (*ATP1B3*^+^ and *GJB2*^+^) (Table 2). 

WNT1 and follicular clusters were mainly present in scalp-derived skin, confirming their role in follicular hair biogenesis, while channel cluster was highly under-represented in all samples. Basal keratinocytes were characterized by the expression of different markers: The basal1 cluster was marked by *CXCL14* and *DMNK*, while the basal2 cluster was marked by *CCL2* and *IL1R2*. Only foreskin biopsies showed a third basal cluster, referred to as basal3, which was enriched for *AREG* expression, an EGFR ligand that promotes keratinocyte proliferation [31]. The relevance of AREG-EGFR signaling in the proliferative context has also been suggested in a single-cell dataset generated from long-term expanded skin [43].

A comparison between adult trunk- (Figure 1C), fetal-(7–10 weeks post-conception), and atopic dermatitis- or psoriasis-derived skin (Figure 2A) was performed by Reynolds et al., who characterized epidermal non immune, dermal non-immune, antigen-presenting, and lymphoid and mast cells groups for a total of 528,253 cells. Among the adult keratinocytes, the authors identified a cluster of undifferentiated (*KRT14*^+^ and *KRT5*^+^), one of differentiated (*KRT1*^+^ and *KRT10*^+^), and one of proliferating cells(*CDK1*^+^ and *PCNA*^+^) (Table 2), as previously reported [25,31]. Fetal-derived keratinocytes differed from the adult ones due to the expression of *KRT8*, *KRT18*, and *KRT19*. Trajectory analysis suggested that the undifferentiated cluster contains SCs, which give rise to proliferating progenitors and differentiated cells [25].

Wang et al. analyzed foreskin keratinocytes (Figure 1A) derived from five healthy donors (17,553 cells), although only one donor (4598 cells) was extensively characterized [33]. Epidermal cells were split into seven clusters, four composed of basal keratinocytes (BAS-I to IV) (around 23%, *KRT14*^+^, *KRT5*^+^, and *CDH3*^+^); two comprising suprabasal spinous (around 54%, *KRT1*^+^, *KRT10*^+^, *DSG1*^+^, and *CDH1*^+^) and granular (around 16%, *DSC1*^+^, *KRT2*^+^, *IVL*^+^, and *TGM3*^+^) cells; and one composed of melanocytes. BAS-I was marked by *PTTG1* and *CDC20*, whilst BAS-II expressed *RRM2*, *HELLS*, *UHRF1*, and *PCLAF* (Table 2). The authors highlighted the role of *HELLS*, *URF1*, and *RRM2* in affecting epidermal homeostasis in organotypic cultures, even if further validations are required to untangle their specific role in epidermis regeneration.

In addition, both basal clusters are enriched for cell-cycle genes (similar to the basal3 cluster from Cheng et al. [31] and the mitotic cluster from Zou et al. [44], later discussed in this review) even though cell-cycle regression was used in the analysis, meaning that differences in their transcriptome are not only strictly related to the cell cycle itself. BAS-III (*ASS*^+^, *COL17A1*^+^, and *POSTN*^+^) and BAS-IV (*GJB2*^+^ and *KRT19*^+^) (Table 2) clusters expressed genes located at the rete ridges level, as confirmed by immunofluorescence analysis [33].

In the study by Zou et al., eyelid-derived skin biopsies from young (18–28 years old), middle-aged (35–48 years old), and old (70–76 years old) donors were analyzed (Table 1, Figure 1K). A total of 9 samples (35,678 cells) were used for subsequent analysis. Among the 11 clusters identified, 5 were composed of keratinocytes divided into basal (BC, 30%, *KRT1*4^+^); mitotic (MC, 11%, *MKI6*7^+^); vellus hair follicles (VHF, 5%, *SOX9*^+^, *KRT6B*^+^, and *SFRP1*^+^), spinous (SC, 49%, *KRT10*^+^) and granular cells (GC, 5%, *FLG*^+^) (Table 2).

Principal component analysis (PCA) of differentially expressed genes (DEGs) indicated that middle-aged keratinocytes were more similar to older rather than younger keratinocytes. Upregulated genes in aged samples were associated with apoptotic and cytokine-mediated signaling, whereas downregulated genes were linked to proliferation, ECM organization, and DNA repair pathways [44]. By increasing cluster resolution, both basal and mitotic clusters were subdivided into three different clusters (BC1 14%, BC2 12%, and BC3 4%; and MC1 7%, MC2 2.5%, and MC3 1%, respectively, on the total keratinocytes). Based on pseudotime and DEG analyses, the authors suggested that cells belonging to the mitotic clusters were the proliferating counterpart of the corresponding basal cluster. Indeed, both BC1 and MC1 expressed *COL17A1*, *IGFBP3*, and *KRT14*. On the other hand, BC3 and MC3 expressed *CYP27A1*, *S100A9*, and *S100A8*, suggesting that these two small clusters were composed of cells committed to a more inflammatory state. BC2 and MC2 clusters, instead, showed *KRT10* expression, which marks the commitment to the differentiation process (Table 2). Further characterization of these differences would be desirable [44]. Nonetheless, this observation would support the classic model of epidermal self-renewal, where SCs give rise to TACs and differentiated cells [52].

This model has been confirmed through pseudotime trajectory analysis, in which basal cells give rise to spinous and granular cells, but no clear consensus exists on the trajectory position of the proliferative cluster compared with the quiescent one [33,35,44].

Wiedemann et al. recently studied the differences in thebehavior of cells derived from the sole, the hip, and the palm (Table 1, Figure 1E–G). Particularly, they investigated the common features and differences among palmoplantar and non-palmoplantar areas [38]. The authors analyzed the transcriptomic profile of 15,423 cells, 9471 of which were keratinocytes. Three clusters were identified as basal (basal I-III) (*KRT14*^+^, *KRT15*, *TP63*^+^, *ITGB1*^+^, and *ITGB4*^+^), two as spinous (*KRT10*^+^, *KRT1*^+^, *SBSN*^+^, and *KRTDAP*^+^) and one as granular (*FLG*^+^, *LOR*^+^, and *SPINK5*^+^) (Table 2). Basal-cell clustering is mostly coherent with the clusters identified in the foreskin by Wang et al. Nonetheless, Wiedemann et al. identified an additional spinous cluster in adult-derived skin, highlighting peculiarities present in the palm, the sole, and the hip. Monocle trajectory analysis confirmed that basal I and II clusters are at the starting point of the pseudotime trajectory, followed by basal III, spinous, and granular clusters [38].

Interestingly, similar skin populations were identified in the long-term expansion (LTE)-derived skin biopsies. LTE is the gold standard procedure for scar reconstruction and reconstructive surgery. This technique allows for a moderate constant skin expansion due to the stretch forces applied to expanders beneath the skin. Four biopsies (two saline-injected expansions and two controls) were analyzed using scRNA-seq (Table 1, Figure 1B,L) [43]. In total, 22,223 cells were analyzed, and keratinocytes were clustered into basal (47.29%, BAS, *KRT14*^+^, *KRT15*^+^, and *COL17A1*^+^); proliferating (6.76%, PC and KI67^+^); spinous (35.58%, SPN, *KRT10*^+^, and *KRT1*^+^); granular (2.24%, GRN, *FLG*^+^, and *IVL*^+^); and hair-follicle clusters (2.43%, HF, *CD59*^+^, and *CD200*^+^) (Table 2). PC cluster was further divided into three subclusters in accordance with Zou et al. No changes in clustering type or differentiation trajectory were reported after expansion therapy, suggesting that a strong proliferating stimulus does not alter epidermal behavior [43].

### 3.2. The Wound-Healing Process Analyzed at Single-Cell Level

Martinengo et al. estimated a global pooled prevalence of 2.21 per 1000 people for chronic wounds of mixed etiologies [53]. In chronic wounds, inflammation, fibrosis, and necrosis coexist, leading to a hampered skin regeneration process. Understanding the complex human wound-healing mechanisms would greatly impact the success of diagnosis and therapy of both acute and chronic wounds. ScRNA-seq data generated from chronic wounds represent a powerful tool, offering new perspectives on the healing process [37,41,54].

Comparisons of chronic non-healing pressure ulcer skin (PU), normal acute wounds (AWs), and skin from matched healthy donors have been performed (Table 1, Figure 2B). A total of 1170 single cells were collected and analyzed using Smart-seq2 technology. In agreement with Cheng et al., keratinocytes were classified into four clusters (KC1-4). KC2 and KC4 were classified as proliferating basal cell clusters: the first was characterized by the expression of genes linked to adhesion and SC signature; the last was characterized by the expression of immunomodulatory genes. The remaining KC1 and KC3 clusters represented the spinous and granular cells, respectively. All the identified clusters were detected in both PU- and AW-derived samples, but changes in their relative abundance were reported. An increase in the percentage of spinous (KC1) and granular (KC3) cells was evident in AW, compared with healthy skin. In contrast, in PU-derived skin, fewer spinous (KC1) and basal (KC2 and KC4) keratinocytes were detected, while a major persistent granular (KC3) cluster was present. Moreover, an increase in the proportion of immune cells was shown in chronic wounds compared with the other samples. Over-representation analysis (ORA) showed that PU keratinocytes overexpressed neutrophil-mediated immunity and apoptosis-related genes [41]. These findings were also confirmed in diabetic foot ulcers, another type of chronic wound, where inflammation and programmed cell death pathways were both upregulated in dermal fibroblasts [50].

In order stratify the patients with PU, the most variable genes on all PU keratinocytes were used to perform PCA. Two groups of patients were identified: t the PU_G2 group was characterized by the enrichment in keratinocytes expressing proliferation-related markers, while keratinocytes from PU_G1 group displayed the upregulation of genes involved in MHC-II-mediated antigen presentation and IFN-γ response. The stimulation of human primary keratinocytes with cell-free wound fluid from PU_G1 patients (but not PU_G2) induced keratinocytes expression of MHC-II-related genes. This effect was blocked by neutralizing IFN-γ in the wound fluids, suggesting that IFN-γ may account for MHC II expression in PU keratinocytes [41].

These different mechanisms in PU highlight the importance of molecular wound diagnosis. Indeed, distinct molecular hallmarks highly correlate with different clinical outcomes that could be exploited in novel personalized medicine strategies concerning chronic wounds. For instance, a targeted molecular diagnosis could highlight the pathogenic mechanisms occurring in a patient’s wound. This could enable the repurposing of anti-IFN-γ antibodies previously developed for other disease treatments, to improve wound healing by blocking IFN-γ pathways [41,55,56].

Another therapeutic opportunity was investigated by Singh et al. Their effort focused on the possibility of a pharmacological reversal of DNA hypermethylation, since mouse model studies showed that this might be a feasible solution to rescue tissue regeneration [37,57]. Epithelial–mesenchymal transition (EMT) is responsible for the initiation of re-epithelialization required for wound closure. The loss of this epithelial plasticity leads to chronic wound persistence. The DNA methylation profile of chronic wound edges (WEs) compared with unwounded samples (UWs) revealed that EMT-related genes and their upstream regulator TP53 were hypermethylated in WEs. ScRNA-seq data analysis from 25,168 cells from chronic WEs and 25,561 cells from UW skin allowed researchers to distinguish two keratinocyte clusters, identified as Kera1 (*KRT14*^+^ and *KRT1*^+^) and Kera2 (*KRT19*^+^ and *KRT7*^+^) (Table 1, Figure 2B) [37]. WE samples were marked by the absence of Kera2 cells, which expressed genes relevant to the cell plasticity and the metabolism switch required during the EMT and the preneoplastic progression [58,59]. Differences also arose in the Kera1 cluster, which displayed a lower expression of TP53 target genes in WE-derived cells, probably due to TP53-promoter hypermethylation, as suggested by methylome data. This finding has been further validated in in vitro studies and mouse models [37]. Thus, the presence of TP53-demethylated locus in a diseased hypermethylated genome paves the way for further investigations. New therapeutic strategies might act on the peculiar epigenetic landscape of the wound-compartmentalized microenvironment.

### 3.3. Inflammatory Skin Diseases: Towards Precision Medicine Using Single-Cell Data

Non-communicable inflammatory skin diseases (ncISDs), such as psoriasis or atopic dermatitis, are a major cause of global disease burden due to their frequency, heterogeneity and complexity. Driven by an intricate interplay of genetics and environmental factors, they are a crucial challenge of modern medicine [60,61]. Due to the lack of models able to predict therapeutic responses, many individuals do not benefit from available therapies [46,60]. Thus, new omic approaches could provide a deep phenotyping of key cell types in ncISDs, offering new information to tackle these diseases [46,51]. Currently, most of the studies are focused on psoriasis (Figure 2C) [25,46,47,51] and atopic dermatitis (AD or eczema) (Figure 2A) [25,40,46,47], while some others also investigate lichen planus (LP) (Figure 2D) [46], erythrokeratodermia variabilis (Figure 2E) [47], lupus erythematosus (LE) (Figure 2F) [42,48], and vitiligo (Figure 2G) [29].

TNFα and IL-17 cytokines contribute to the dysregulation of the immune response in keratinocyte-driven rashes through NF-kB activation. The systemic blockade of these cytokine pathways is beneficial, but a specific targeting in the accessible skin tissue would reduce systemic side effects. A20 is a promising targetable NF-kB-inhibiting partner protein. The scRNA-seq analysis of 42,105 cells showed that A20 overexpression inhibited the expression of an inflammatory-genesignature upregulated in psoriasis, AD, and erythrokeratodermia variabilis samples. Based on this finding, the A20 in vivo upregulation could represent a therapeutic path to dampen skin inflammation in a variety of ncISDs [47].

The scRNA-seq analysis of psoriasis, atopic dermatitis, and healthy control biopsies enabled the identification of four clusters of undifferentiated, differentiated, proliferating, and inflammatory differentiated cells. The inflammatory-differentiated cluster, expressing inflammation markers (*ICAM1*, *TNF*, and *CCL20*), as well as low levels of undifferentiation-related (*TP63* and *ITGA6*) and differentiation-related (*KRT1* and *KRT10*) markers, was expanded in psoriasis skin [25]. Previous reports have shown that psoriasis lesional skin was enriched in inflammatory-differentiated keratinocytes and mitotic cells, suggesting an increase in the cell plasticity in disease states [31].

He et al. provided a further single-cell characterization of AD biopsies (Table 1). AD keratinocytes were characterized by the upregulation of epidermal proliferation-associated genes, including *S100* and protease inhibitors *SERPINB4*, consistent with epidermal hyperplasia. The lesional proliferating and suprabasal keratinocytes from their dataset overexpressed *KRT6*, *KRT6A*, and *KRT16*, which are usually enriched in hyperproliferative and wound-healing states [40]. Similar results were confirmed via a scRNA-seq analysis of suction blistering and punch biopsy of both healthy and AD skin [30] (Table 1).

Cutaneous lupus erythematosus (CLE) has been studied by means of scRNA-seq and spatial transcriptomics approaches [42]. Biopsies were collected in both lesional and non-lesional areas from 7 CLE and 14 healthy control skin samples (Table 1, Figure 2F). Among the 25,675 cells, common clusters were composed of basal (*KRT15*^+^, *COL17A1*^+^, *DST*^+^, *KRT14*^+^, *POSTN*^+^, *CXCL14*^+^, *S100A2*^+^, *KRT5*^+^, *SYT8*^+^, *CYR1*^+^), spinous (*KRT10*^+^, *LY6D*^+^, *KRTC*^+^, and *KRT1*^+^); granular or supraspinous (*FLG*^+^, *LOR*^+^, *SLURP1*^+^, *FLG2*^+^, *C1orf68*^+^, *HOPX+*, *CNFN*^+^, *SPINK5*^+^, *CALML5*^+^, *CDSN*^+^, and *KRT10*^+^); follicular (*GJB6*^+^, *KRT6B*^+^, *TM4SF1*^+^, *GJB2*^+^, *CHCHD10*^+^, *CRABP2*^+^, *WFDC3*^+^, *S100P*^+^, *MUCL1*^+^, and *KRT17*^+^); and cycling (*STMN1*^+^, *CENPF*^+^, *TUBA1B*^+^, *PTTG1*^+^, *HMGB2*^+^, *NUSAP1*^+^, *TOP2A*^+^, *TK1*^+^, *MKI67*^+^, and *HIST1H4C*^+^) cells (Table 2) [42]. Basal and spinous subclusters were enriched in lesional CLE-derived keratinocytes. Nearly all the cells belonging to these clusters showed high IFNα-signaling responses. Surprisingly, the upregulation of cytokine-related pathways was also pronounced in non-lesional skin from CLE patients, suggesting the existence of a prelesional state. These data were corroborated through the spatial transcriptomic analysis of healthy and CLE-derived biopsies, also providing one of the first reports of spatial localization of these clusters in healthy skin [42]. Similar results were obtained in lupus nephritis (LN), a major organ manifestation of systemic lupus erythematosus, which could lead to acute or chronic renal failure [48]. An IFN-response signature was detectable in the tubular cells of LN patients. Similar results were obtained in non-lesional skin biopsies from the same individual, suggesting that the scRNA-seq analysis of readily accessible skin biopsies could be exploited in clinics to reflect kidney injury [48].

Notably, ncISDs (particularly LP, AD, and psoriasis) have also been studied through spatial transcriptomics using Visium technology by 10× Genomics. From each group of patients, a lesional and non-lesional section was withdrawn [46] (Table 1). This assay is not formally at single-cell resolution, as it generally covers between 1 and 10 cells per spot. Therefore, the authors did not directly analyze single keratinocytes but implemented a density-based clustering method. This allowed them to correlate cytokine expression to responder gene signatures, according to spatial features, paving the way for curative treatment strategies, such as antigen-specific immunotherapies [46], as already suggested for pemphigus vulgaris [62,63].

Although all these strategies might prove successful, further investigations of cell–cell interaction in lesional and non-lesional samples are required, as they might shed light on the subsequent processes that trigger inflammation and lesion formation [42,48]. The identification of the specific disease-causing antigens, and the subsequent targeting of cytokine-producing immune cells in the inflammatory microenvironment, will speed up the development of ncISD-tailored therapies [46].

## 4. Single-Cell Molecular Profiling of In Vitro Cultured Human Primary Keratinocytes

Regenerative therapies to restore damaged or diseased epithelium are routinely applied in clinics. Several patients have been successfully treated, completely saving or changing the lives of those affected by extensive burns, skin genetic diseases, and burned-cornea blindness [7,9,24]. The success of all these different approaches relies on the possibility of in vitro culture epithelial SCs. In 1975, H. Green was able to overcome the limitations previously observed in the cultivation of epidermal cells in surface cultures by coculturing them with mitotically inactivated murine fibroblasts [64]. This protocol has been used since 1984 to treat burned patients using autologous cultivated epidermal graft [7].

Given their potency and plasticity, in vitro cultured keratinocytes have been characterized over time. Already in 1987, H. Green was able to describe three clonal types: holoclones, meroclones, and paraclones [65]. Holoclone-forming cells displayed the highest proliferative potential and unique self-renewal capacity; therefore, they are considered epithelial SCs. Conversely, meroclone- and paraclone-forming cells showed less proliferative potential and are normally referred to as TACs. As progenitors, they lack self-renewal capacity and progressively lose their proliferative potential, giving rise to terminally differentiated cells. These assumptions were initially based on clinical observations, in which permanent epithelial regeneration was only possible when an adequate number of holoclone-forming cells were present in the grafted culture [66]. Nonetheless, the formal proof that holoclone-forming cells are SCs only became possible in 2017, when a patient affected by JEB was successfully treated using corrected autologous keratinocytes. Clonal tracing analysis has been performed using the provirus integration site as a unique clonal marker, allowing researchers to track cell progeny in the transduced grafted skin [10].

To better understand the biology on which these functional differences rely, the molecular characterization of the three clonal types has been carried out. TP63, YAP, and FOXM1 were found to be the crucial transcription factors (TFs) that sustain epithelial SC self-renewal [36,67,68,69]. The microarray analysis of holoclone-, meroclone-, and paraclone-derived progenies identified a list of 526 genes differentially expressed in holoclones, as compared to meroclones and paraclones, hence defined as holoclone signature [36].

To gain insight into clone-founding cells, scRNA-seq was applied to the clinical-grade culture of human primary keratinocytes extracted from two healthy donors’ truncal skin biopsies (Table 1, Figure 1C) [23]

The transcriptomic profiles of 7354 cells were analyzed. Three clusters expressed high levels of clonogenic/basal markers (*KRT14*^+^, *TP63*^+^, *ITGA6*^+^, and *ITGB1*^+^), whereas differentiated cells, identified as clusters TD1 and TD2, expressed *SERPINB3*, *SFN*, *KRT10*, *IVL*, and *SPINK5*. Among the three clonogenic clusters, the one expressing the holoclone signature to the highest extent was named H, containing holoclone-forming cells. M and P clusters expressed lower levels of holoclone signature and contained meroclones and paraclone-forming cells. The H cluster expressed high levels of genes linked to cell cycle, DNA repair, microtubule organization, and YAP and FOXM1 signaling pathways. Monocle analysis confirmed that the H cluster is the starting point of a unique and mainly linear trajectory that proceeds through M and P clusters, giving rise to TD1 and TD2 cells. Single-cell data were further validated using gain- and loss-of-function experiments, confirming the crucial role of FOXM1 as a downstream target of YAP to sustain SC self-renewal potential during in vitro culture [36,67].

The presence of an adequate number of SCs in the graft is one of the main concerns during translational phases [70]. Jayarajan et al. performed a single-cell analysis to study the anoikis-preventing effect of Rho-associated kinase inhibitor (ROCKi) in maintaining keratinocyte SC self-renewal (Figure 1K, Table 1) [45,71].

ROCKi enhanced cell clonogenic potential after a 6-day treatment. In line with Enzo et al., they identified clusters of SCs, TACs, and differentiated cells in untreated conditions. Surprisingly, they showed a ROCKi-driven reduction in the percentage of holoclone-forming cells, reversible upon withdrawal. The single-cell profiling of long-term-treated keratinocytes could shed light on molecular mechanisms responsible for SC reduction, possibly confirming previously published data [72,73].

The epigenetic profiling of in vitro cultured keratinocytes at the single-cell level was published by Khavary’s lab. To study the gene network controlling cell fate, they developed perturb-ATAC (assay for transposase-accessible chromatin) (Table 1, Figure 1A), a method able to measure the impact of CRISPR modification on chromatin accessibility in each cell [32]. To this aim, undifferentiated and calcium-induced differentiated keratinocytes were subjected to single-cell ATAC sequencing (scATAC-seq). This allowed the alignment of single cells onto a unique and mainly linearly pseudotime differentiation trajectory. The differentiation was driven by 67 TFs that clustered in 3 modules, which were differentially activated along the differentiation route and were able to recapitulate (i) the proliferation state and progenitors’ mitosis control; (ii) the mid-differentiation and the cell–cell adhesion; and (iii) the late keratinization in terminally differentiated cells. The authors identified TFs required for the differentiation program and examined epigenetic changes upon single or combined gene silencing of those TFs. This method could ideally identify co-regulated regulatory elements, providing insights for further biological validations [32].

## 5. Network Identification Using Single-Cell Dataset

Studies involving cell–cell interactions might play a pivotal role in understanding physiological and pathological tissue conditions. Starting from single-cell data, different approaches have been used to investigate such synergy in the human epidermis.

Network studies have emerged from the re-analysis of Cheng’s dataset, through which a network of transcription factors modules controlling self-renewal and differentiation has been established. However, the biological validation of these results was only partial [74].

Notably, scRNA-seq data provide novel opportunities to study receptor–ligand interactions, identifying networks of communicating cells in tissues. In a previous study by Wang et al., cell–cell interactions were determined using the SoptSC algorithm. They predicted the interactions occurring via WNT, JAK-STAT, NOTCH, and TGF-β signaling pathways. The WNT pathway was active in most of the basal and spinous cells, while TGF-β signaling was restricted to the basal layer. The upper layers of the epidermis were enriched in NOTCH4 and JAK/STAT signaling [33].

Other research groups studied cell–cell interactions in healthy and inflamed skin. Starting from scRNA-seq data, the differential expression of ligand–receptor pairs in AD versus healthy controls was analyzed [40]. *CCL2* was significantly upregulated in lesional AD versus control basal keratinocytes. The gene was also abundantly expressed by a unique population of AD *COL6A5*^+^ and *COL18A1*^+^ inflammatory fibroblasts, unrecognized by previous single-cell studies on healthy dermal fibroblasts [75]. CCL2 receptors (*CCR1* and *CCR2*) were expressed on macrophages and dendritic cells. Another chemokine, *CCL27*, was upregulated in lesional keratinocytes, not only in basal but also in suprabasal clusters. *CCL27* upregulation suggests the establishment of signaling via CCR10 in T cells. Taken together, these findings highlight the interactions between immune and other cell types in skin [40].

### Bioinformatic Tools for Cell Network Analysis

Starting from sc-RNA seq data, several user-friendly tools have been recently developed (Table 3). Here, we report those employed to study cell–cell interactions in healthy, wounded, and cancerous skin.

NicheNet, developed in 2019, enables the assessment of not only ligand–receptor interactions but also its predicted impact on intracellular signaling. As a result, it can predict which ligands influence gene expression in another cell, which target genes are affected by each ligand, and which signaling mediators may be involved [76]. Ji et al. applied NicheNet to study keratinocyte-predicted ligands that modulate the transcriptomic profile of squamous cell carcinoma (SCC) microenvironment-specific cells at the leading edge (Figure 2H). The tumor-specific keratinocyte signaling with nearby cancer-associated fibroblasts was mediated by several receptor–ligand pairs, such as MMP9-LRP1 and TNC-SDC1. The tool also confirmed that tumor keratinocytes at the leading edge resembled an EMT-like population, as they expressed TGFB1 and integrins such as ITGA3 and ITGB1 [34].

The database CellPhone DB, released in 2020 [77], considers the subunit architecture of both ligands and receptors, accurately representing heteromeric complexes. Solè-Boldo et al. used CellPhone DB to study the interactions between fibroblasts and other skin cell types during aging (Table 1, Figure 1H). The analysis showed that the number of interactions between fibroblasts and undifferentiated keratinocytes decreased during aging [39].

Released in 2021, the CellChat database considers the additional effects that soluble and membrane-bound stimulatory and inhibitory cofactors exert on these interactions. This tool was employed by Sun et al. to identify intercellular communications driving skin regeneration for long-term expansion therapy [43]. Ligand–receptor pairs AREG-EGFR, CD96-NECTIN, and LAMIN-CD44 were identified as the most significantly upregulated signaling. These pathways might be essential for mechanical stretching and likely contribute to the maintenance of long-term skin regeneration. As expected, the EGF pathway resulted in upregulation in the expanded skin as well [43]. In the work of Guerrero-Juarez et al., CellChat allowed the description of the interactions between fibroblasts and the basal cell carcinoma (BCC) stroma in affecting tumor growth. Fibroblast-secreted WNT5A was identified as the ligand interacting with receptors FZD6 and FZD7 expressed by the basal layer’s keratinocytes [49]. Yakupu et al. took advantage of CellChat to provide new insights into PU cellular connections, using data from Li et al. [41,54]. This analysis revealed that intercellular communication is enhanced in both number and strength, mostly between spinous keratinocytes and other clusters in PU, compared with AW samples. Nonetheless, a comparison with healthy skin is missing. The authors suggested that, in PU, spinous and mitotic cells mainly receive signals from melanocytes. In particular, the protease-activated receptor (PAR) signaling pathway mediated by the CTSG-F2RL1 ligand-receptor pair is one of the most activated in PU [54]. CTSG is a serine protease mainly expressed by melanocytes and involved in inflammation, while F2RL1 is widely expressed by fibroblasts and keratinocytes, where it plays an important role as a driver of inflammatory response. These data suggest that PARs might represent an important therapeutic target [54,78].

In 2023, Cang et al. developed a new tool called COMMOT that added spatial information to the study of cell–cell interactions. To validate it, they applied the algorithm to a wide range of datasets, demonstrating that it can consistently capture cell–cell communications (CCCs) already described in the literature [79]. COMMOT was used to study the role of CCCs in human epidermal development. Starting from receptor–ligand pairs annotated in CellChat DB, COMMOT predicted that the molecular interactions between GAS6 and PROS1 with TYRO3 were significant in granular cells and moderately present in basal cells. This notion was confirmed using IF and in situ hybridization (ISH) analysis. Furthermore, the authors analyzed four signaling pathways with well-established roles in epidermal homeostasis, namely WNT, TGF-β, NOTCH, and JAK/STAT. All the signaling cascades were mainly upward-directed, from the basal to suprabasal layers. Conversely, some signals were downward-directed toward the basal layers at the bottom of the ridges [79,80].

## 6. Conclusions

Single-cell techniques have revolutionized the field of biology, offering unprecedented insights into the complexity and heterogeneity of biological systems. Undoubtedly, the ability to profile individual cells has greatly expanded the understating of tissue organization and dynamics, unraveling the mechanisms that underlie normal development and disease progression. Furthermore, the development of multiomic approaches facilitates the description of the genomic and gene expression profile within each cell, providing a multifaceted and comprehensive understanding of cellular behavior.

However, it is important to be cautious of potential pitfalls. Considering the enormous amount of information derived from these techniques, standardizing bioinformatic analysis is crucial for the generation of comparable data. Misleading interpretations could also arise from sample-handling procedures, as they directly influence cell collection and stress-related cellular responses. Therefore, in silico findings need careful experimental validation to avoid computational and sample-handling-derived artifacts.

Skin is a complex tissue, and its cellular heterogeneity is now being tackled by single-cell analysis, in both physiological and pathological conditions. The ability to characterize the epidermis architecture at the single-cell level, combined with cell–cell communication models, offers powerful insights into its many layers of biological complexity.

Despite the potentialities of these emerging tools, some biological questions remain unanswered. Indeed, further analyses are required to elucidate which pathways are involved in in vivo self-renewal and symmetric/asymmetric cell division. A starting point in deciphering epidermal stem-cell molecular profile involves in vitro experiments, in which they are activated in a condition mimicking wound healing.

The importance of unraveling skin complexity is linked to the possibility of (i) monitoring clinically relevant cell populations for advanced therapy (e.g., epidermal SCs in cell and gene therapy applications); (ii) defining a patient-specific diagnosis; and (iii) providing targets to improve precision medicine.

Excitingly, new frontiers are emerging in the field of single-cell analysis. Among them is the possibility to deconvolute cell complexity at the protein level, which promises an even more accurate characterization of biological processes.

## Figures and Tables

**Table 2 ijms-24-08544-t002:** Schematic description of cell clusters and corresponding markers (italic) obtained after bioinformatic analysis of the transcriptomic profile of single cells collected from human healthy skin biopsies; arrow: results from subclustering analysis; n/a: not available.

Article	Undifferentiated/Basal	Differentiated		Proliferating	Hair Follicle-Like/Others
Reynolds et al. Science. 2021 [25]	Undifferentiated KC: *KRT14*, *KRT5*	Differentiated KC: *KRT1*, *KRT10*		Proliferating: *CDK1*, *PCNA*	
Cheng et al. Cell Rep. 2018 [31]	Basal: *KRT14*, *KRT5*	Spn: *KRT1*, *KRT10*	Grn: *LOR*, *FLG*	Mitotic: *CCND1*, *PCNA*	Follicular: *MGST1*, *S100A2*
	→ Basal1: *CXCL14*, *DMNK*				Channel: *ATP1B3*, *GJB2*
	→ Basal2: *CCL2, IL1R2*				Wnt1: *SFRP1*, *FRZB*
	→ Basal3: *AREG*				
Wang et al. Nat Commun. 2020 [33]	BAS: *KRT14*, *KRT5 CDH3*	SPN: *KRT1*, *KRT10*, *SBSN*, *KRTDAP*	GRN: *DSC1*, *KRT2*, *IVL*, *TGM3*		
	→ BAS-I: *PTTG1, CDC20*				
	→ BAS-II: *RRM2, HELLS, UHRF1, PCLAF*				
	→ BAS-III: *RRM2, HELLS, UHRF1, PCLAF*				
	→ BAS-IV: *GJB2, KRT19*				
Wiedemann et al. Cell Rep. 2023 [38]	Undifferentiated keratinocytes: *KRT14*, *KRT5*, *TP63*, *ITGB1*, *ITGB4*	Differentiated keratinocytes: *KRT1*, *KRT10*, *SBSN*, *KRTDAP*	Terminally differentiated keratinocytes: *FLG*, *LOR*, *SPINK5*		
	→ Basal1: n/a	→ Spinous 1: *GRHL3, FOSL1, SOX9*			
	→ Basal2: n/a	→ Spinous 2: *FOS*, *GADD45B*			
	→ Basal3: n/a				
Solè-Boldo et al. Commun Biol. 2020 [39]	Keratinocytes undiff: *KRT5*, *KRT14*, *TP63*, *ITGA6*, *ITGB1*	Keratinocytes diff.: *KRT1*, *KRT10*, *SBSN*, *KRTDAP*			
Billi et al. Sci Transl Med.2022 [42]	Basal: *KRT15*, *COL17A1*, *DST*, *KRT14*, *POSTN*, *CXCL14*, *S100A2*, *KRT5*, *SYT8*, *CYR61*	Spinous: *KRT1*, *LY6D*, *KRT6C*, *KRT16*	Supraspinous: *FLG*, *LOR*, *SLURP1*, *FLG2*, *C1orf68*, *HOPX*, *CNFN*, *SPINK5*, *CALML5*, *CDSN*	Cycling: *STMN1*, *CENPF*, *TUBA1B*, *PTTG1*, *HMGB2*, *NUSAP1*, *TOP2A*, *TK1*, *MKI67*, *HIST1H4C*	Follicular: *GJB6*, *KRT6B*, *TM4SF1*, *GJB+*, *CHCHD10*, *CRABP2*, *WFDC3*, *S100P*, *MUCL1*, *KRT17*
Sun et al. Front Cell Dev Biol. 2022 [43]	Basal cells (BAS): *KRT14*, *KRT15*, *COL17A1*	Spinous cells (SPN): *KRT10*, *KRT1*	Granular cells (GRN): *FLG*, *IVL*	Proliferative cells (PC): *MKI67*	Hair follicle (HF): *CD59*, *CD200*
				→ PC1: *KRT1*, *KRT10*, *DMKN*	
				→ PC2: *KRT15*, *COL17A1*, *IGFBP3*, *KRT14*	
				→ PC3: *CALML3*, *PTN*, *CYP27A1*	
Zou et al. Dev Cell. 2021 [44]	Basal cells (BC): *KRT14*	Spinous cells (SC): *KRT10*, *KRT1*	Granular cells (GC): *KRT10*, *KRT1*, *FLG*	Mitotic cells (MC): *KRT14*, *KRT5*, *KI67*, *TK1*	Vellus Hair Follicul (VHF): *SOX9*, *KRT6B*, *SFRP1*
	→ BC1: *COL17A1, IGFBP3, KRT14*			→ MC1: *COL17A1, IGFBP3, KRT14*	
	→ BC2: *POSTN, MYC, ID1*			→ MC2: *KRT1, KRT10, DMNK*	
	→ BC3: *CYP27A1, S100A9, S100A8*			→ MC3: *CYP27A1, S100A9, S100A8*	

**Table 3 ijms-24-08544-t003:** List of bioinformatic tools for cell network analysis. Provided links have been accessed on 6 May 2023.

Tool	Year of Release	Link	Availability	
			Online	Upon Installation
NicheNet	2019	https://www.nichenet.be		x
CellPhone DB	2020	http://www.cellphonedb.org	x	
CellChat	2021	http://www.cellchat.org	x	
COMMOT	2023	https://github.com/zcang/COMMOT		x

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
