# Peer review of "Decoding the Human Epidermal Complexity at Single-Cell Resolution"

_ijms, 2023, doi:10.3390/ijms24108544_

Round 1

Reviewer 1 Report

The submitted work is messed up. Information needs to be better organized.

Lines 25-27 have no reference;

Line 39 quotes an author “H. Green” because there are other cited references than this one (4,5);

Line 49: there is a dash at the end of the sentence;

Lines 58-59: It's meaningless. It seems that the reasoning has been cut off.

Lines 64-65: I don't understand what the authors mean by making this access available online.

Line 83 and line 91: Reference 20 was skipped, that is, it was not quoted in the order of the text;

Table 1 is not formatted correctly and does not appear in the numerical order of the references. It is important that authors use the instructions for authors available on the journal's page;

Line 259 and 260. References are not numbered correctly;

Line 289-290 has no reference;

Line 322 reference 50 is not in numerical order. Section “4.2 The wound healing process analyzed at single cell level” reference 50 was cited 5 times. It was pointless to elaborate a section with a large repetition of this reference.

Line 519: Sentence too short. It was meaningless.

The article is confusing. The information needs to be better organized for a publication.

English is confused.

The submitted work is messed up. Information needs to be better organized.

Lines 25-27 have no reference;

Line 39 quotes an author “H. Green” because there are other cited references than this one (4,5);

Line 49: there is a dash at the end of the sentence;

Lines 58-59: It's meaningless. It seems that the reasoning has been cut off.

Lines 64-65: I don't understand what the authors mean by making this access available online.

Line 83 and line 91: Reference 20 was skipped, that is, it was not quoted in the order of the text;

Table 1 is not formatted correctly and does not appear in the numerical order of the references. It is important that authors use the instructions for authors available on the journal's page;

Line 259 and 260. References are not numbered correctly;

Line 289-290 has no reference;

Line 322 reference 50 is not in numerical order. Section “4.2 The wound healing process analyzed at single cell level” reference 50 was cited 5 times. It was pointless to elaborate a section with a large repetition of this reference.

Line 519: Sentence too short. It was meaningless.

The article is confusing. The information needs to be better organized for a publication.

English is confused.

Reviewer 2 Report

In this review, Polito et al. describe with great precision and hindsight, the studies of single cell RNA Seq carried out on the skin and more particularly on the epidermis and its keratinocyte precursors. It is particularly appreciable that this review begins with a chapter on "Skin biopsies processing" because the technical approach of single cell separation influences the analyzes of the results obtained.

The review is very well written even if chapters 3 and 4 could have been reversed to start with the tissue and continue on the keratinocytes in culture.

The fact that authors claim and document precisely the differences in studies on the location and age of the tissues studied is fundamental and must be hightligh as done.

However, few minor points can be considered:

- For ease of reading, it would be preferable that Table 1 should be represented in a "landscape" format with a larger character size.

- Page 4 and elsewhere, it should be discussed that, as usual, only highly expressed genes are highlighted in described clusters. What about low expressed genes in specific cluster. For example, a down regulation of a specific Transcription Factor in a specific cluster (in keratinocyte Stem Cell cluster, for example) could be extremely relevant for this specific biological function.

- Figure 3 is relatively disappointing because it only presents a series of markers already known for several years. It should be possible to highlight the new markers of the different subpopulations discovered in several studies, in the basal layer, for example.

- Page 6, line 256: "URF1" should be replace by "UHRF1"

- Even if melanocytes are important cell constituant of epidermis, this chapter 4.4, only support by two references (...), is not essential for this type of review and should be deleted. This chapter could be replace by another, presenting scRNA Seq on epidermal carcinomas.

- Links to acces to bioinformatics tools (NicheNet and COMMOT) should be added. Indicate, for each one if software are free available or not (perhaps by adding a short table summarizing all tools).

This manuscript should be revised with minor changes

Round 2

Reviewer 1 Report

The authors made substantial changes to the article, but for some reason the article was misformatted. There are repeated figures and blank pages. This needs to be reviewed. Tables are not formatted according to the instructions for authors.

Authors should review the instructions for authors available on the journal page.

Minor editing of English language required.

Author Response

We are glad that reviewer 1 appreciates our new manuscript and highlithed only formatting issues.

  • Regarding the misformatted text, unfortunately we do not see any repeated figures and blank pages, then I cannot fix this point.
  • Regarding the table issue, this is very general and we try to follow the editorial guidelines as previously suggested. Could the reviewer provide more detailed point on what is not following the editorial roles?